# Open-Surface Water Bodies Dynamics Analysis in the Tarim River Basin (North-Western China), Based on Google Earth Engine Cloud Platform

**Jiahao Chen** [1,2] , **Tingting Kang** [1,2], **Shuai Yang** [1,2], **Jingyi Bu** [1,2] , **Kexin Cao** [1,2] **and Yanchun Gao** [1,*]

1. Key Laboratory of Water Cycle and Related Land Surface Processes, Institute of Geographical Sciences and Natural Resources Research, Chinese Academy of Sciences, Beijing 100101, China; chenjiahao18@mails.ucas.ac.cn (J.C.); kangtt.14b@igsnrr.ac.cn (T.K.); yangs.17b@igsnrr.ac.cn (S.Y.); bujy.16b@igsnrr.ac.cn (J.B.); caokx.18s@igsnrr.ac.cn (K.C.)
2. College of Resources and Environment, University of the Chinese Academy of Sciences, Beijing 100049, China
* Correspondence: gaoyanc@igsnrr.ac.cn; Tel.: +86-010-6488-8991

**Abstract:** The Tarim River Basin (TRB), located in an arid region, is facing the challenge of increasing water pressure and uncertain impacts of climate change. Many water body identification methods have achieved good results in different application scenarios, but only a few for arid areas. An arid region water detection rule (ARWDR) was proposed by combining vegetation index and water index. Taking computing advantages of the Google Earth Engine (GEE) cloud platform, 56,284 Landsat 5/7/8 optical images in the TRB were used to detect open-surface water bodies and generated a 30-m annual water frequency map from 1992 to 2019. The interannual changes and trends of the water body area were analyzed and the impacts of climatic and anthropogenic drivers on open-surface water body area dynamics were examined. The results show that: (1) ARWDR is suitable for long-term and large-scale water body identification, especially suitable for arid areas lacking vegetation. (2) The permanent water area was 2093.63 km$^2$ and the seasonal water area was 44,242.80 km$^2$, accounting for 4.52% and 95.48% of the total open-surface water area of he TRB, respectively. (3) From 1992 to 2019, the permanent and seasonal water bodies of the TRB all showed an increasing trend, with obvious spatial heterogeneity. (4) Among the effects of human activities and climate change, precipitation has the largest impact on the water area, which can explain 65.3% of the change of water body area. Our findings provide valuable information for the entire TRB's open-surface water resources planning and management.

**Keywords:** open-surface water bodies; Landsat image; Google Earth Engine; Tarim River Basin; climate change

## 1. Introduction

Open-surface water bodies, including lakes, rivers, streams, reservoirs, and ponds, are extremely important water resources for agriculture, aquaculture, industrial production, aquatic, and terrestrial ecosystems [1]. As a type of land cover, open-surface water bodies play an important role in climate regulation, biogeochemical cycles, and surface energy balance [2]. Climate change and increased climate variability can strongly impact open-surface water resources [3–5], causing significant intra- and inter-annual water bodies area changes [6,7]. Due to climate change and human activities, permanent water bodies have been reduced by nearly 90,000 km$^2$, 2.02% of the all water bodies from 1984 to 2015 [8]. In recent decades, many countries, especially developing countries, have experienced

a rapid urbanization process. Changes in surface water caused by human activities strongly affect surface temperature, soil moisture, biodiversity, and ecosystem functions [9]. Therefore, monitoring the dynamic changes of open-surface water is of great significance to the health of the natural environment and the sustainable development of the social economy [10].

Remote sensing technology has the advantages of wide coverage, instantaneous imaging, high spatial resolution, short revisit period, and fast data acquisition, and it is now widely used in the identification of surface water bodies and their change detection [10–21] on a large scale. Common remote sensing data sources used to detect open-surface water bodies include Moderate Resolution Imaging Spectrometer (MODIS) images [10–14], Landsat images [15–19], Sentinel images [20,21], etc. However, traditional open-surface water bodies studies, due to the difficulty of processing large amounts of satellite imageries [22], usually use fewer imageries of large scale areas at a particular period. In recent years, cloud-based high-performance data computing platforms such as Google Earth Engine (GEE) [23], NASA Earth Exchange (NEX), Data Cubes, etc. [24] have been developed rapidly. Among them, GEE was widely used in various studies [25], such as on drought monitoring [26], land use/land cover [27], fire monitoring [28], mine mapping [29], evapotranspiration [30], city [31], rice [32], wetland changes [33], vegetation cover [34], and open-surface water body detection [8,22,35]. For example, Pekel et al. [8] published a dataset of water bodies changes based on the GEE platform for global large scale long series; Deng et al. [36] analyzed the characteristics of the spatiotemporal changes of surface water bodies in the Yangtze River Basin from 1984 to 2018; Wang et al. [22] studied the spatiotemporal changes of surface water bodies in the middle reaches of the Yangtze River based on GEE; and Wang et al. [35] used GEE to conduct a meticulous study on the long-term sequence dynamics of Poyang Lake. Thus, GEE provides a new perspective for studying the dynamics of long time series of open-surface water bodies.

Current water bodies extraction algorithms mainly include water bodies recognition algorithms based on band combination (single-band threshold method [37], multi-band spectral relationship method [38], water body index and threshold method [39–42]), and machine learning classification algorithms (random forest [35], support vector machine [43], etc.). Among them, the most prevalent is the water body index method based on band combination, such as Tasseled Cap Wetness (TCW) [44], Normalized Difference Water Index (NDWI) [41], Modified Normalized Difference Water Index (MNDWI) [42], Water Index 2015 (WI2015) [40], Automated Water Extraction Index (AWEI) [39], Revised Normalized Difference Water Index (RNDWI) [45], and Enhanced Water Index (EWI) [46]. Beeri et al. [47] combined three water indexes, Sum457, ND5723, and ND571, and proposed a new algorithm SNN. Sum457 was developed by Al-Khudhairy et al. [48]. ND5723 and ND571 can further reduce the effects of aerosols and other atmospheric substances. The water index method looks for the strongest and weakest water bodies characteristic reflection bands in the multi-spectral image and expands with the help of proportional calculation. The water index method has obvious advantages in suppressing disturbance factors such as vegetation, shadow, soil, etc. but is too dependent on the strength of the ground feature relationship. When the land feature relationship weakens, the water body index method cannot obtain satisfactory results. Because the use of a single water body detection index may confuse water bodies and vegetation, combining the vegetation index can improve the accuracy of water bodies identification. Menarguez [49], based on the Land Surface Water Index (LSWI), NDWI, MNDWI, Normalized Difference Vegetation Index (NDVI) [50], and Enhanced Vegetation Index (EVI) [51], proposed a water bodies extraction algorithm, and their results show that this integrated method is more sensitive to water bodies signals. On this basis, Zou et al. [52] proposed a new water detection rule: when a pixel meets the conditions EVI < 0.1 and MNDWI > NDVI or MNDWI > EVI, it is regarded as water. This detection rule has been applied to different scales and different types of water detection tasks in the United States [53], Oklahoma [52], Mongolian Plateau [54], and Poyang Lake [35] and has achieved good results. This water detection rule effectively solves the problem of confusion between vegetation and water bodies [52], which is suitable for water body detection tasks in wet areas with lush vegetation, but some pixels with a vegetation index less

than the water body index will also be mistakenly identified as water, especially in the arid area with sparse vegetation. Thus, this water body detection rule needs to be modified to make it more suitable in the arid area.

The Tarim River is the longest inland river in China, the fifth-largest inland river in the world and is a typical arid-zone inland river. In the past 60 years, due to the intensive human economic and social activities in the TRB, the natural ecosystems have been severely damaged, resulting in a series of environmental problems. Such as 321 km of downstream flow disruption, drying up of lakes, a significant decline of groundwater level, extensive decay of desert riparian forests, intensification of desertification process, and damage to biodiversity [55], which have affected the economic and social development of the oasis in the basin and the survival of the people. Thus, it is necessary to explore the temporal and spatial dynamic changes of the open-surface water bodies in the TRB. However, most previous studies on the dynamic changes of open-surface water bodies in the TRB were limited to the analysis of individual years and individual regions [56,57], and there were some deviations in the results. Therefore, we used the big data advantage of the GEE platform to analyze the changes of open-surface water bodies in the TRB by using all available historical images.

The objectives of this study were: (1) to propose a new water detection rule for the arid region water detection task; (2) to analyze the long-term changes of open-surface water bodies in the TRB from 1986 to 2019 based on the GEE platform and all available Landsat imagery; and (3) to analyze the causes of changes of open-surface water bodies in the TRB. Besides, the study can help the understanding of long-term changes in the open-surface water bodies in the TRB and promote the development of water resources management in the TRB.

## 2. Materials and Methods

### 2.1. Study Area

The TRB is located in northwestern China (73°10′–96°38′ E and 34°55′–43°48′ N) (Figure 1). It is the largest inland river basin in China, with an area of $1.085 \times 10^6$ km$^2$, about 11.3% of the total area of China. The TRB has a complex geographical environment, with the Tianshan Mountains in the north, the Kunlun Mountains in the south and west, and plains and deserts in the middle and east [58] (Figure 1a). Affected by the geographical location and terrain, TRB belongs to the typical temperate arid continental climate, which has little rain and strong evaporation. The average annual precipitation is 124.08 mm, and it is concentrated in the western and northern mountainous areas (Figure 1b). The evaporation capacity is 1800–2900 mm. The average annual temperature is 10.7 °C, and the highest temperature in the year is 39–42 °C [59], The temperature presents a spatial law of decreasing from the interior of the basin to the marginal mountainous area (Figure 1c).

In this study, the TRB is divided into nine sub-basins: the Hotan River Basin (HT), the Yarkant River Basin (YK), the Kashgar River Basin (KS), the Aksu River Basin (AKS), the Weikan-Kaidu-Peacock River Basin (WKP), Rivers in Southern Xinjiang (SR), the Tarim Main Stream (TMS), the Taklamakan Desert (TD), and the Kumtag Desert (KD).

Following clockwise, HT originates from the mountains in the southwest and merges into the Tarim River. YK originates from the mountains in the west and flows east to the Tarim Basin. KS is located in the northwest of the TRB and does not flow into the mainstream. AKS is the largest source of water for the Tarim River [60], which is located in the northwest of the Tarim Basin. WKP is located in the northern TRB, originating from the Tianshan Mountains, including the Weigan River Basin, the Kaidu River Basin, and the Peacock River Basin. SR is on the southern edge of the Tarim Basin, including some small river basins such as that of the Qarqan River. TMS does not produce its flow [60], relying entirely on the source of water supply; its starting point is located at the confluence of the Yarkant River, Aksu River, and Hotan River and belongs to Lake Tetema. TD is located in the center of the Tarim Basin, the largest desert in China, and the second-largest flowing desert in

the world [61], with extremely scarce precipitation. KD is located in the eastern part of the Turpan Basin, where precipitation is scarce too.

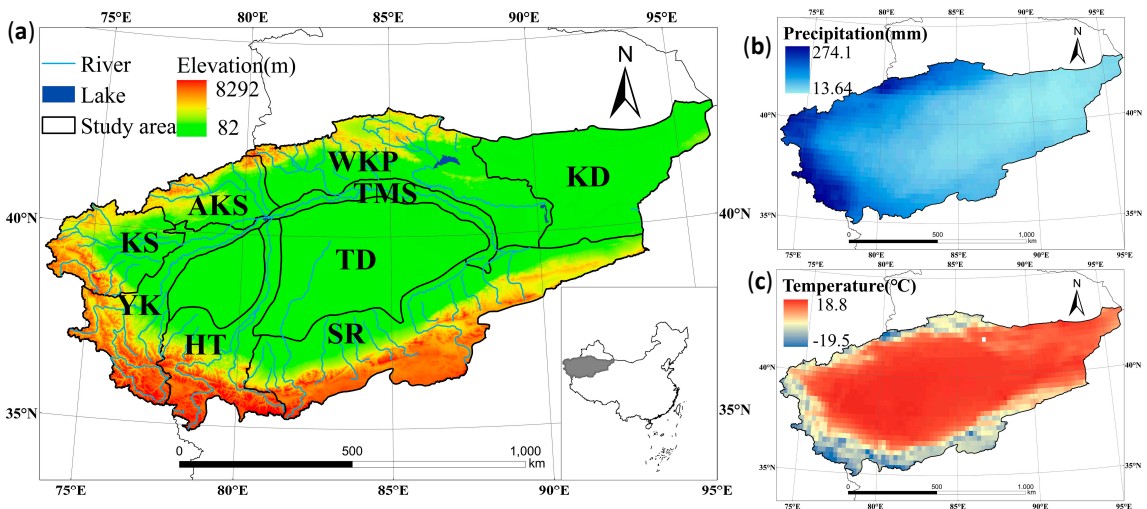

**Figure 1.** The DEM (**a**); precipitation (**b**); and temperature (**c**) in 2019 of the TRB. HT, Hotan River Basin; YK, the Yarkant River Basin; KS, Kashgar River Basin; AKS, Aksu River Basin; WKP, Weikan-Kaidu-Peacock River Basin; SR, Rivers in Southern Xinjiang; TMS, the Tarim Main Stream; TD, Taklamakan Desert; KD, Kumtag Desert.

## 2.2. Data

This study collected all available Landsat 5, 7, and 8 surface reflectance images of the TRB from 1986 (Landsat 5 was launched in March 1984, but it did not image the TRB during 1984–1985) to 2019. Among them, the Landsat 5 and 7 surface reflectance datasets were generated by processing the Landsat ecosystem disturbance adaptive processing system (LEDAPS) algorithm, while the Landsat 8 surface reflectance datasets were generated by processing the Landsat surface reflectance code (LaSRC) algorithm [62]. In this study, six spectral bands, including blue band, green band, red band, near-infrared band, short-wave infrared 1 band, and short-wave infrared 2 band, were selected from the surface reflectance image data of Landsat 5, 7, and 8 for the extraction of water bodies. Landsat 5 covered TRB for the first time in 1986 and terminated in 2012; Landsat 7 was launched in 1999, and the scan line error occurred in 2003, but it continues to operate to this day; and Landsat 8 was launched in 2013 and has been operating well until now. In total, 56,284 images from the Landsat series were used in this study, including 19,771 Landsat 5 during 1986–2012; 25,060 Landsat 7 during 1999–2019; and 11,453 Landsat 8 during 2013–2019. The spatial distribution, temporal distribution, and month distribution of total observation counts from 1986 to 2019 are presented in Figure 2.

Moreover, to analyze the relationship between water bodies area and climate factors (precipitation, temperature, and evapotranspiration), PERSIANN-CDR daily precipitation data [63] and 3-h Global Land Data Assimilation System (GLDAS) product [60] from 1984 to 2019 were selected. The Joint Research Centre (JRC) global surface water dataset [8] was used to filter the obvious classification errors (e.g., mountain shadow) in the water bodies extraction results. Besides, SRTM DEM data were used to demonstrate the elevation of the TRB.

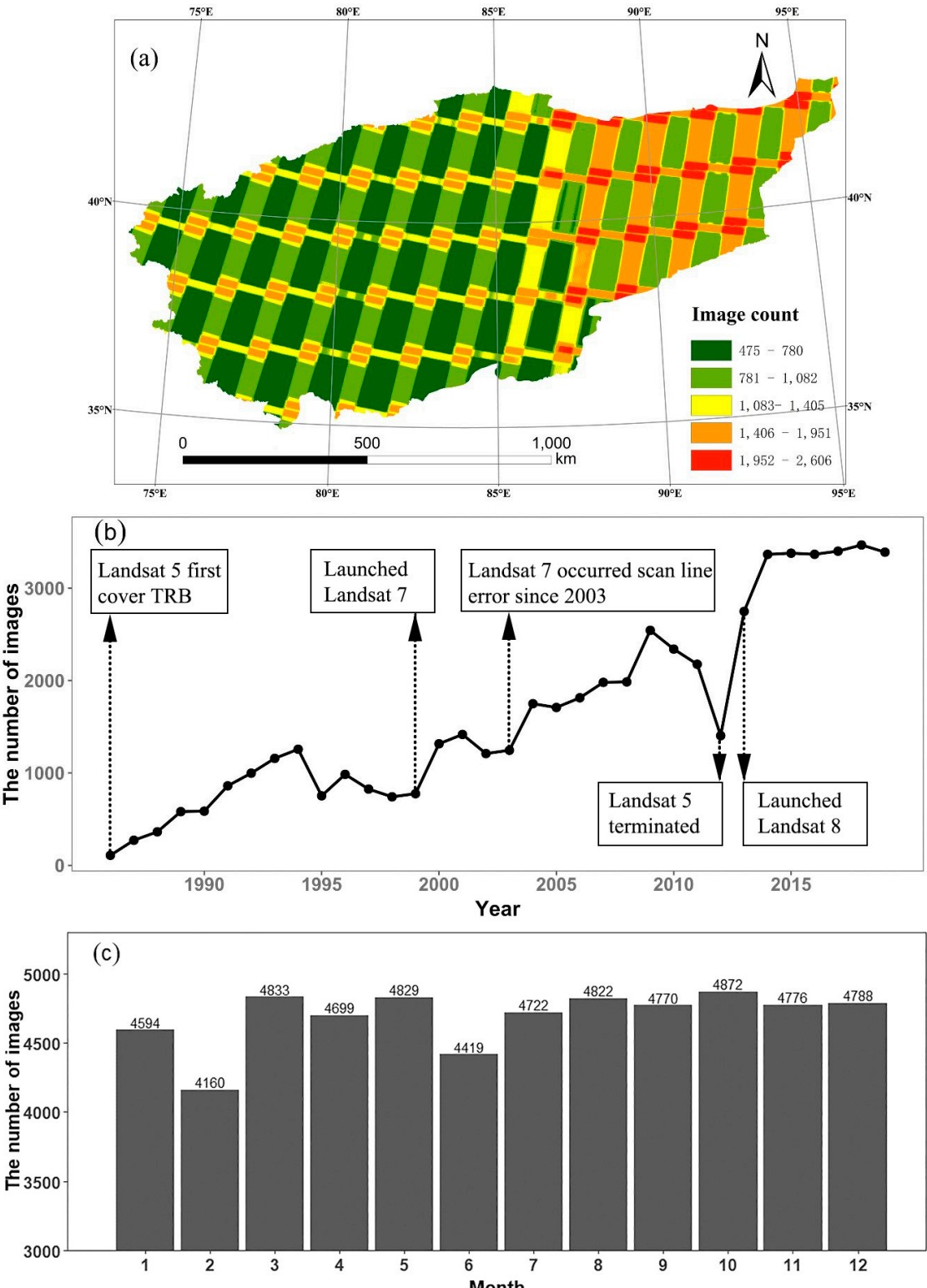

**Figure 2.** The Landsat image of the study area: (**a**) the spatial distribution, (**b**) temporal distribution, and (**c**) seasonal distribution.

### 2.3. Methods

#### 2.3.1. Original Water Detection Rule

To reduce the problem of commission errors in the mixed pixels of water bodies and other land cover types, especially vegetation on wet surfaces [64] in MNDWI, Zou et al. [52] combined MNDWI and vegetation indices (NDVI and EVI) to weaken the influence of vegetation on the water map algorithm. Specifically, only pixels with stronger water signals than vegetation signals (MNDWI > NDVI or MNDWI > EVI) are detected. To further remove the noise caused by vegetation, EVI is used to eliminate vegetation wetland pixels (EVI < 0.1). Therefore, only those pixels that meet the rule MNDWI > NDVI or MNDWI > EVI and EVI < 0.1 are classified as open-surface water pixels. The remaining pixels are classified as non-water pixels.

#### 2.3.2. Arid Region Water Detection Rule

Although Zou et al.'s [52] rule has been widely used and achieved good results [22,35,53], we found that this water detection rule does not perform well in the TRB. Some pixel points in the bare ground meet the above detection rules, but their visual interpretation results are not water bodies, because many pixels in the arid region meet the vegetation index is smaller than the water index, but their water index is also small or even negative. For further research on the causes of misclassification, we randomly selected 50 water samples and 50 samples that were misidentified as water by Zou et al.'s [52] water detection rule and obtained the average spectral curves (Figure 3) of two different land types. As shown in Figure 3, the spectra of the two types of samples are very different, the values of each band of the misidentified sample are higher than the water sample, the reflectance of the water sample increases from the blue band to the green band and then continuously decreases, the reflectance of the sample misidentified as water continuously increases from the blue band to the red band, and the reflectance of the red and near-infrared bands is approximately the same and continuously decreases. Although the spectral curves of the two types of samples were significantly different, both meet the water detection rules for EVI < 0.1 and MNDWI > NDVI or MNDWI > EVI.

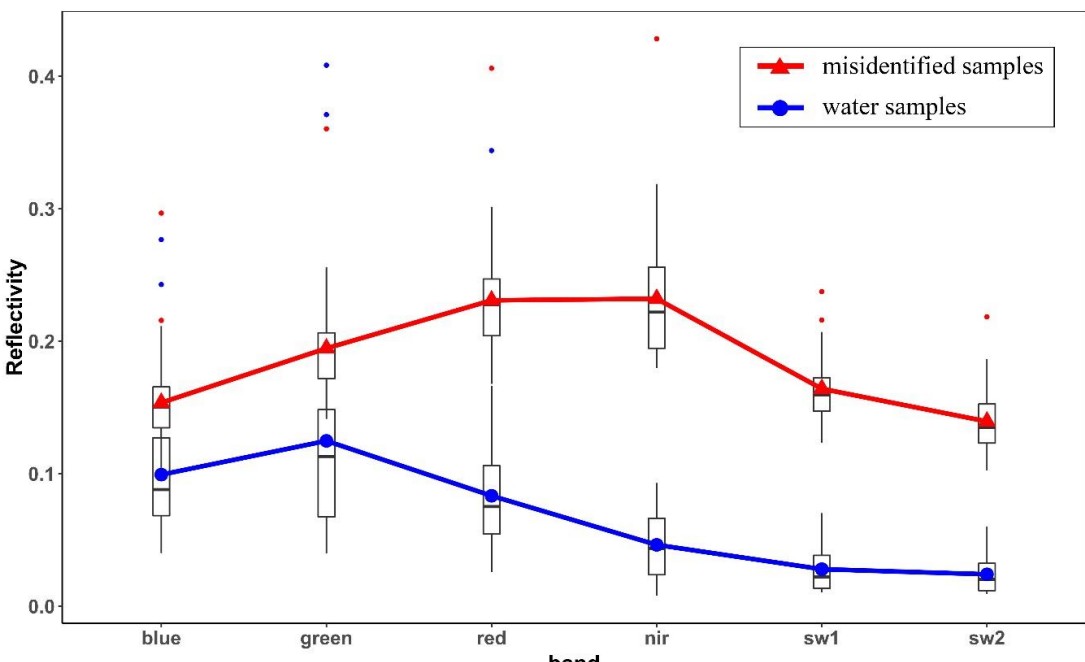

**Figure 3.** Spectra of water samples and misidentified samples.

To improve the accuracy of water detection in arid regions such as the TRB, a new water detection rule, the Arid Region Water Detection Rule (ARWDR), is proposed, which adds the limit of water

index to the original water detection rule. After calculating MNDWI for these 50 water samples and 50 misidentified samples, the mean value of MNDWI of water samples was 0.595, and the variance was 0.03; the mean value of MNDWI of misidentified samples was 0.079, and the variance was 0.008. Therefore, adding the restriction of MNDWI to the original water body identification rules improved the recognition accuracy. Further research found that the mean value of the NDWI value of the misclassified sample points with MNDWI > 0.1 was −0.103, and the variance was 0.005; therefore, the restriction of NDWI > −0.1 can be added to further improve accuracy. Finally, we obtained the arid region water detection rule: NDWI > −0.1, MNDWI > 0.1, EVI < 0.1, and MNDWI > NDVI or MNDWI > EVI. To compare the advantages and disadvantages of ARWDR with other water detection methods, this study further compared ARWDR with 10 other existing methods (Table 1).

**Table 1.** Water index list.

| Method | Index | Threshold |
|---|---|---|
| MNDWI [42] | $MNDWI = (B_{green} - B_{SWIR-1})/(B_{green} + B_{SWIR-1})$ | MNDWI > 0 |
| WI2015 [40] | $WI2015 = 1.7204 + 171B_{green} + 3B_{red} - 70B_{Nir} - 44B_{SWIR-1} - 71B_{SWIR-2}$ | WI2015 > 0 |
| AWEI$_{nsh}$ [39] | $AWEI_{nsh} = 4 \times (B_{green} - B_{SWIR-1}) - (0.25 \times B_{Nir} + 2.75 \times B_{SWIR-1})$ | AWEI$_{nsh}$ > 0 |
| RNDWI [45] | $RNDWI = (B_{Sir} - B_{red})/(B_{Sir} + B_{red})$ | RNDWI > 0 |
| EWI [46] | $EWI = (B_{green} - B_{Nir} - B_{Mir})/(B_{green} + B_{Nir} + B_{Mir})$ | EWI > 0 |
| SNN [47] | $Sum457 = B_{Nir} + B_{SWIR-1} + B_{SWIR-2}$ $ND5723 = [(B_{SWIR-1} + B_{SWIR-2}) - (B_{green} + B_{red})]/[(B_{SWIR-1} + B_{SWIR-2}) + (B_{green} + B_{red})]$ $ND571 = [(B_{SWIR-1} + B_{SWIR-2}) - B_{blue}]/[(B_{SWIR-1} + B_{SWIR-2}) + B_{blue}]$ | (Sum457 < 0.188) or (ND5723 < −0.457) or (ND571 < 0.04) or (Sum457 < 0.269 and ND5723 < −0.234 and ND571 < 0.40) |
| NDWI+ VI [49] | $EVI = 2.5 \times (B_{Nir} - B_{red})/(B_{Nir} + 6.0 \times B_{red} - 7.5 \times B_{blue} + 1)$ $NDVI = (B_{Nir} - B_{red})/(B_{Nir} + B_{red})$ $NDWI = (B_{green} - B_{Nir})/(B_{green} + B_{Nir})$ | EVI < 0.1 and (NDWI > NDVI or NDWI > EVI) |
| MNDWI + VI [52] | $EVI = 2.5 \times (B_{Nir} - B_{red})/(B_{Nir} + 6.0 \times B_{red} - 7.5 \times B_{blue} + 1)$ $NDVI = (B_{Nir} - B_{red})/(B_{Nir} + B_{red})$ $MNDWI = (B_{green} - B_{SWIR-1})/(B_{green} + B_{SWIR-1})$ | EVI < 0.1 and (MNDWI > NDVI or MNDWI > EVI) |
| LSWI + VI [49] | $EVI = 2.5 \times (B_{Nir} - B_{red})/(B_{Nir} + 6.0 \times B_{red} - 7.5 \times B_{blue} + 1)$ $NDVI = (B_{Nir} - B_{red})/(B_{Nir} + B_{red})$ $LSWI = (B_{Nir} - B_{SWIR-1})/(B_{Nir} + B_{SWIR-1})$ | EVI < 0.1 and (LSWI > NDVI or LSWI > EVI) |
| AWEI + VI [36] | $EVI = 2.5 \times (B_{Nir} - B_{red})/(B_{Nir} + 6.0 \times B_{red} - 7.5 \times B_{blue} + 1)$ $NDVI = (B_{Nir} - B_{red})/(B_{Nir} + B_{red})$ $AWEI_{sh} = B_{blue} + 2.5 \times B_{green} - 1.5 \times (B_{Nir} + B_{SWIR-1}) - 0.25 \times B_{SWIR-2}$ $AWEI_{nsh} = 4 \times (B_{green} - B_{SWIR-1}) - (0.25 \times B_{Nir} + 2.75 \times B_{SWIR-1})$ | (AWEI$_{nsh}$ − AWEI$_{sh}$ > −0.1) and ((MNDWI > EVI) or (MNDWI > NDVI)) |
| ARWDR | $EVI = 2.5 \times (B_{Nir} - B_{red})/(B_{Nir} + 6.0 \times B_{red} - 7.5 \times B_{blue} + 1)$ $NDVI = (B_{Nir} - B_{red})/(B_{Nir} + B_{red})$ $MNDWI = (B_{green} - B_{SWIR-1})/(B_{green} + B_{SWIR-1})$ $NDWI = (B_{green} - B_{Nir})/(B_{green} + B_{Nir})$ | (NDWI > −0.1) and (MNDWI > 0.1) and (EVI < 0.1) and ((MNDWI > EVI) or (MNDWI > NDVI)) |

### 2.3.3. Water Bodies Extraction Process

This study was based on the GEE platform for the detection of open-surface water bodies in the TRB. First, clouds, cloud shadows, and ice were masked according to the quality band of Landsat surface reflectance products generated from the Fmask algorithm [65] (pixels with scan line errors were also masked from the Landsat 7 image starting in 2003), and the remaining pixels were considered

good observations for the next analysis. Each image was then processed using ARWDR to obtain a preliminary water body distribution map. Mountain shadows and other classification errors were finally removed from the JRC global surface water bodies distribution dataset [8,36]. The flow chart of this study is shown in Figure 4.

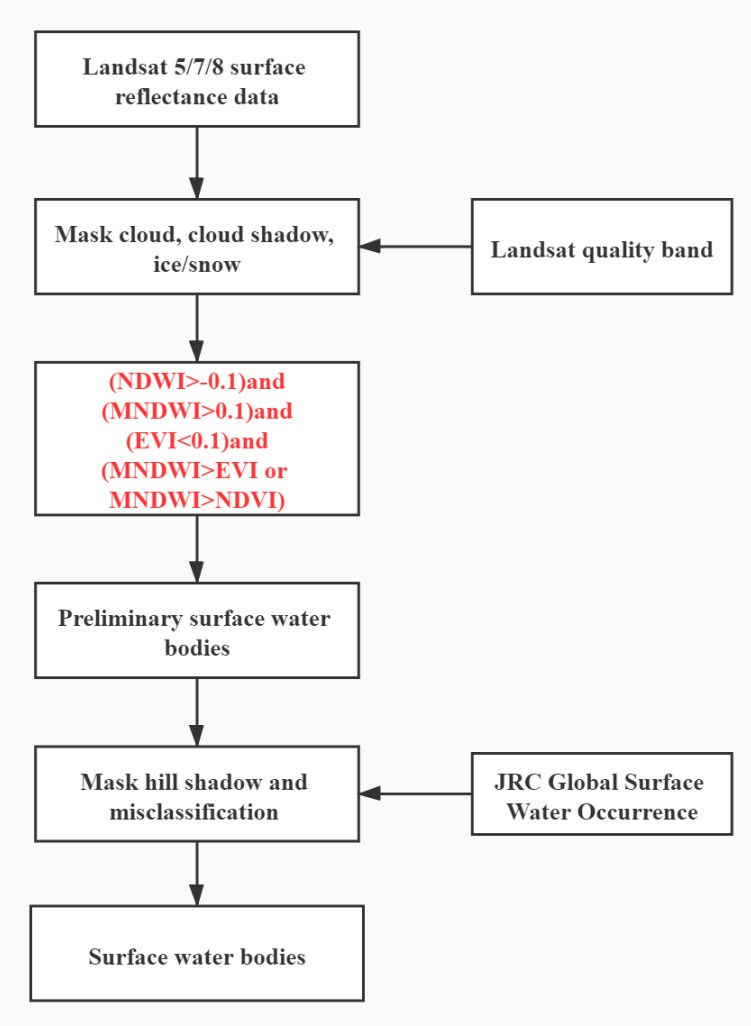

**Figure 4.** The flowchart of the extraction of open-surface water bodies in the TRB.

### 2.3.4. Change Analysis of Open-surface Water Bodies

The percentage of the number of times a pixel is identified as water in all good observations of the pixel can be defined as the water inundation frequency (WIF) [36], which is calculated as follows:

$$\text{WIF} = \frac{\sum_{i=1}^{N} w}{N} \times 100\% \tag{1}$$

where $N$ represents the number of good observations in a specific period, and w is a binary variable: when $w = 1$, the pixel is water, and, when $w = 0$, the pixel is non-water. WIF is taken in the range of [0%, 100%]. In previous studies [22,36,52], 75% and 25% were used as thresholds for classifying permanent and seasonal water bodies: the pixel was considered a permanent water bodies when WIF > 75%, a seasonal water bodies when 75% ≥ WIF > 25%, and an invalid water bodies when WIF < 25%. In this way, the uncertainty of image data quality flags and other small probability problems in image preprocessing are reduced. However, considering that the JRC Global Surface Water Occurrence Layer was used in this study to mask mountain shadows and other classification errors and most of

the water bodies in the TRB's WIF are below 0.25, this threshold needs to be modified; thus, 75% and 1% were used as thresholds to classify permanent water bodies and seasonal water bodies. When WIF > 75%, the pixel is considered as permanent water bodies; when 75% ≥ WIF > 1%, it is considered as seasonal water bodies; and, when WIF ≤ 1%, it is considered as an invalid water body. In addition, a WIF greater than 1% and less than or equal to 100% is considered the maximum body of water.

Besides, this study used the Theil–Sen slope estimation method [66] to evaluate the long-term change trend of open-surface water bodies. The formula for calculating the inclination is:

$$\beta = Median\left[\frac{x_j - x_i}{j - i}\right] \tag{2}$$

where $j > i$. $\beta > 0$ indicates that the series has an upward trend; $\beta = 0$ indicates that the series has no trend; and $\beta < 0$ indicates that the series has a downward trend.

## 3. Results

### 3.1. Accuracy Comparison of Different Water Indexes

To compare the accuracy of ARWDR and other water body identification methods on TRB, six Landsat tiles with less than 1% cloud cover in the TRB were selected for this study. they cover various land types and typical water bodies in the TRB, including Lake Bosten, Lake Tetema, Daxihaizi Reservoir, water bodies in Kashgar city, water bodies around Aksu farmland, and alpine lakes near Altun Mountain, corresponding to permanent water bodies, seasonal water bodies, artificial water bodies, urban water bodies, farmland water bodies, and mountain water bodies, respectively.

Each image was de-clouded by the Fmask algorithm [65], and then six bands of the Landsat 5, 7, and 8 surface reflectance image data, including blue band, green band, red band, near-infrared band, short-wave infrared 1 band, and short-wave infrared 2 band, were selected. We randomly generated 60,000 sample points on GEE and the type-stable sample points were preserved by visual interpretation, resulting in a total of 50,817 sample points, including 4907 water bodies sample points and 45,910 non-water sample points.

Comparing the eleven methods with the samples obtained by visual interpretation, the overall accuracy and kappa coefficients of the different methods are shown in Figure 5. It can be seen that the overall accuracy and kappa coefficient of ARWDR are the highest, reaching 0.9972 and 0.9842, respectively; thus, this study used ARWDR for water detection in TRB.

The reason is single water index method, used by, e.g., AWEI$_{nsh}$, EWI, RNDWI, WI_2015, MNDWI, etc., has obvious advantages in suppressing interference factors such as vegetation, shadows, and soil but is overdependent on the strength of the ground feature relationship. When the land feature relationship weakens, the water body index method cannot obtain satisfactory results. Water body identification rules combined with vegetation index, such as NDWI + VI, MNDWI + VI, LSWI + VI, AWEI + VI, etc., use the relative value relationship between the vegetation index and the water body index to effectively distinguish the water body from the vegetation. However, the water body identification rules combined with the vegetation index only pay attention to the relative value relationship between the vegetation index and the water index and ignore the limitations of the absolute value of the water index. Therefore, in arid areas with sparse vegetation or no vegetation coverage, some non-water pixels' water index will be small, but still larger than the vegetation index, and those pixels will be mistakenly identified as water. Therefore, for arid areas such as the TRB, combining two water body identification schemes, using the relative relationship between the vegetation index and the water index, and adding the absolute value limit of the water index, can further improve the accuracy of water body identification; therefore, the recognition accuracy of ARWDR is higher than that of the other ten water recognition methods.

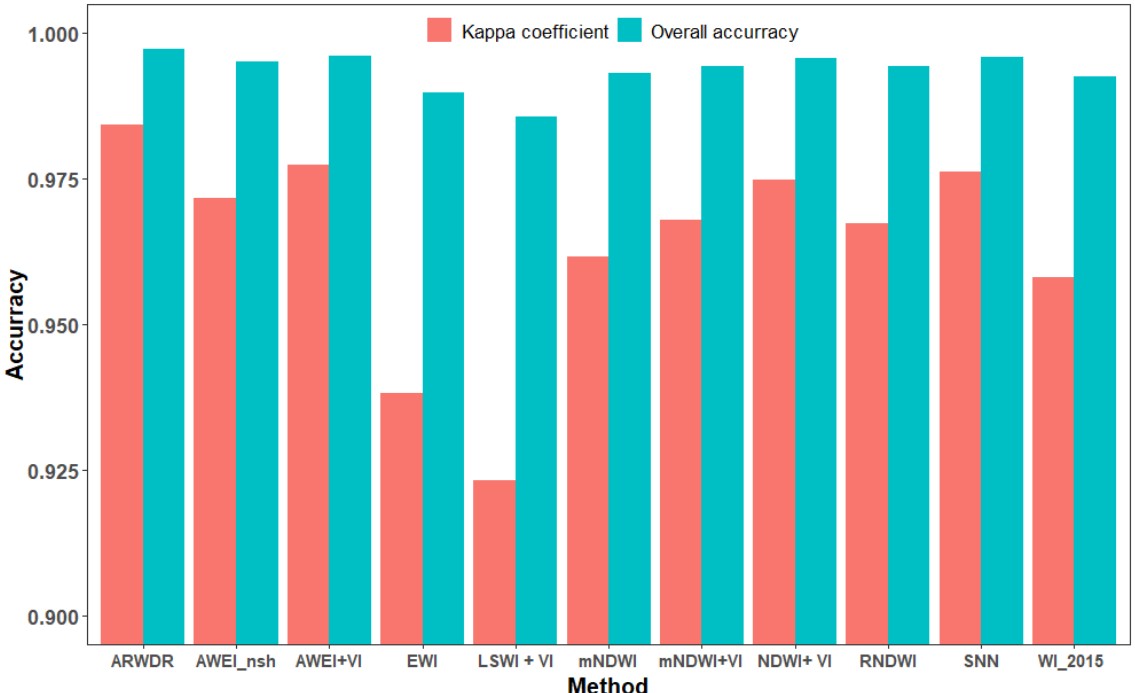

**Figure 5.** Comparison of different water detection rules.

*3.2. Spatial Distribution of Open-Surface Water Bodies in the TRB*

The distribution of open-surface water bodies in the TRB is highly spatially heterogeneous (Figure 6 and Table 2). About 52.65% of the permanent water bodies are distributed in the WKP, followed by 20.87% of the permanent water bodies in the YK and less than 10% of the permanent water bodies in the remaining seven regions. About 24.88% of the seasonal water bodies are located in the WKP, followed by 16.27% of the seasonal water bodies in the SR. About 23.55% of the largest water bodies are located in the WKP, followed by SR with 13.78% of the largest water bodies.

In general, the open-surface water bodies of the TRB are mostly distributed in the piedmont plains and mountainous areas, and there are barely permanent water bodies in the deserts inside the basin. This is because the TRB is far away from the ocean and surrounded by high mountains; therefore, it is difficult for humid gases to enter the basin, and the climate is dry and evaporation is strong. Precipitation is mostly distributed in mountainous areas (Figure 1b), so that surface water is formed in mountainous areas. Runoff from precipitation and meltwater from mountain glaciers flow out of mountainous areas, forming rivers in vast piedmont plains and forming lakes (e.g., Bosten Lake) in low-lying places, or it flows into the mainstream of the Tarim River and then into the tail (e.g., Taitma Lake). In the nine sub-basins, Bosten Lake is located at the lowest point in the local elevation and it receives most of the runoff of the Kaidu River, therefore WKP has the largest permanent water body. Because precipitation in the TRB is mainly distributed in the mountainous areas of YK. Therefore, YK has the largest runoff among the sub-basins, its annual average runoff is $68.75 \times 10^8$ m$^3$ [67], and its open-surface water area is second only after WKP. Since the four sub-basins of KS, HT, SR, and AKS all include some mountainous areas, their water areas are not much different, and they are all smaller than WKP and YK. TMS receives runoff from WKP, YK, KS, HT, SR, and AKS [59], but does not include mountain catchments, therefore its water body area is only greater than those of TD and KD. Since TD and KD are almost desert areas, there are few water bodies in the two sub-basins. However, due to the construction of the Lop Nur potash plant in KD, the permanent water body has increased by 1.52 km$^2$, and the seasonal water body has increased by 372.65 km$^2$.

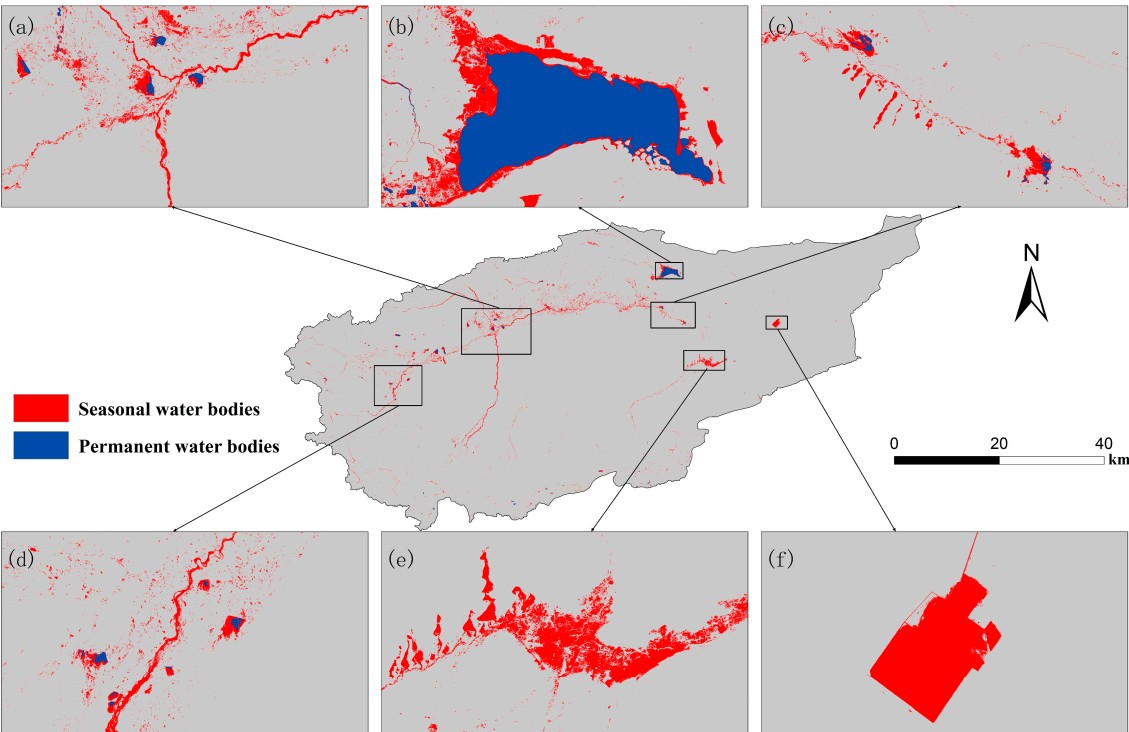

**Figure 6.** The spatial distribution of open-surface water bodies from 1986 to 2019 in the TRB: (**a**) the source of the mainstream of the Tarim River, (**b**) Bosten Lake, (**c**) Daxihaizi Reservoir, (**d**) Yarkant River, (**e**) Tetema Lake, and (**f**) Lop Nor Potash Plant.

**Table 2.** The area of open-surface water bodies from 1986 to 2019 in the TRB. HT, Hotan River Basin; YK, the Yarkant River Basin; KS, Kashgar River Basin; AKS, Aksu River Basin; WKP, Weikan-Kaidu-Peacock River Basin; SR, Rivers in Southern Xinjiang; TMS, the Tarim Main Stream; TD, Taklamakan Desert; and KD, Kumtag Desert.

| Zone | Permanent Water Bodies (km²) | Seasonal Water Bodies (km²) | Max Water Bodies (km²) |
|------|------------------------------|------------------------------|------------------------|
| HT | 128.05 | 4162.69 | 6609.00 |
| YK | 436.92 | 6048.67 | 12,041.59 |
| KS | 159.06 | 5418.37 | 11,455.65 |
| AKS | 99.95 | 4875.39 | 9309.38 |
| WKP | 1102.26 | 11,006.33 | 21,000.30 |
| SR | 105.26 | 7198.40 | 12,675.96 |
| TMS | 62.13 | 4663.80 | 12,183.94 |
| TD | 0 | 248.05 | 1415.35 |
| KD | 0 | 621.14 | 2186.54 |
| Total | 2093.63 | 44,242.80 | 88,877.70 |

This research has identified the open-surface water bodies of TRB from the perspective of remote sensing, calculated the submerged frequency per pixel according to ARWDR, and distinguished permanent water bodies, seasonal water bodies, and non-water. However, water bodies, such as lakes, rivers, and reservoirs, are not classified from the perspective of hydrology. Therefore, we selected six representative water bodies (Figure 6a–f) to analyze the functionality for natural or anthropogenic environments. We also calculated the long-term dynamic changes in the area of these six typical water bodies (Table S1) and created dynamic maps of the historical changes of these six typical water bodies (Figure S1a–f). Figure 6a is the source of the mainstream of the Tarim River and is also an important cotton and wheat production area [68]. In addition to several rivers, there are also reservoirs such as

Shengli Reservoir. These water bodies provide irrigation water for agricultural production in this area. Figure 6b is Bosten Lake, which is the main water source of Korla and undertakes the task of ecological water delivery to the lower reaches of the Tarim River [69]. Figure 6c is a series of reservoirs headed by the Daxihaizi Reservoir. The Daxihaizi Reservoir was built in 1960 to irrigate local crops but caused some side effects such as water cuts in the downstream. Therefore, it has completely withdrawn from agricultural irrigation since 2012 and assumed the task of ecological water delivery to the lower reaches of the Tarim River. Figure 6d is a part of YK, which has the largest agricultural irrigation area in Xinjiang, with high-quality cotton and commercial grain base, and provides plenty of irrigation water for agricultural production in this area [70]. Figure 6e shows Tetema Lake, the tail of the Tarim River. Due to ecological water transportation, Lake Tetema, which had been dry for nearly 30 years, was restored to its surface. This is of great significance to the downstream ecological restoration and the improvement of the ecological vulnerability of the tail Lake area. More importantly, the restoration of Tetema Lake protected the natural green corridors of the lower reaches of the Tarim River [71], preventing TD and KD from converging. The wetland area of Tetema Lake has reached 511 km$^2$, and the vegetation coverage increased significantly [72]. Figure 6f shows the Lop Nur Potash Plant. It is a huge artificial water body built to alleviate the shortage of potash fertilizer in China. It will promote the economic development of Xinjiang, relieve the pressure of China's import of potash fertilizer, and promote the development of China's agriculture to a higher level [73].

### 3.3. Monthly Changes in Open-Surface Water Bodies in the TRB

The open-surface water bodies in the TRB showed a clear seasonal difference, with the area of open-surface water bodies decreasing rapidly from January to April, decreasing slowly from April to July, reaching its lowest value in July, rising slowly from July to October, and rising rapidly from October to December in a "single-valley" trend. The largest area of permanent and seasonal water bodies in January was 4487.97 and 67,809.51 km$^2$, respectively, and the smallest are of permanent and seasonal water bodies in July was 1691.16 km$^2$ and 10,566.24 km$^2$, respectively (Figure 7). This is due to the gradual rebound of temperature in the TRB from January to April, the evapotranspiration increasing, the plants gradually entering the rejuvenation period, and the moisture required for growth reaching maximum, which led to the reduction of open-surface water bodies area. During April–July, temperatures rise to their maximum for the whole year, evapotranspiration of surface water bodies increases to the maximum, but the melting of snow in the mountains gives the river water recharge so that the reduction of open-surface water bodies reaches a relatively calm state. During August–December, temperatures gradually decrease, evapotranspiration decreases, the moisture required for plant growth decreases, and the open-surface water bodies area increases [74].

We can find similar characteristics of monthly variation in the nine sub-basins of the TRB (Figure 8), that is, the area of water bodies was large in winter and small in summer. The maximum surface water bodies area of each sub-basin appears in January, while the minimum surface water bodies area varies from sub-basin to sub-basin, with the minimum water bodies area of KD appearing in May, in June in TD and YK, in July in TMS, HT, and WKP, and in August in RS, AKS, and KS. This may be due to the different climate characteristics and planting structure of each sub-basin. For example, unlike other sub-basins, the minimum water area of TMS does not gradually increase from August to December but experiences another valley value in October (Figure 8g). This is because TMS is the main grain-producing area with a large area of winter wheat, and October is the winter wheat irrigation period [75], which leads to a decrease in the open-surface water area.

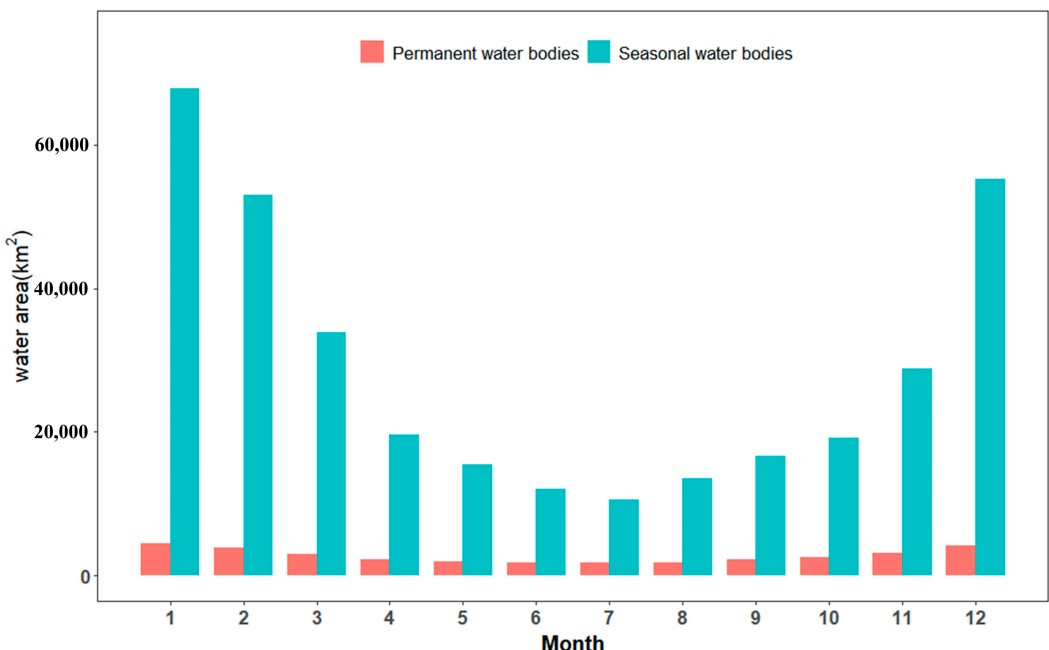

**Figure 7.** The monthly variation of surface water bodies for 1984 to 2019 in the TRB.

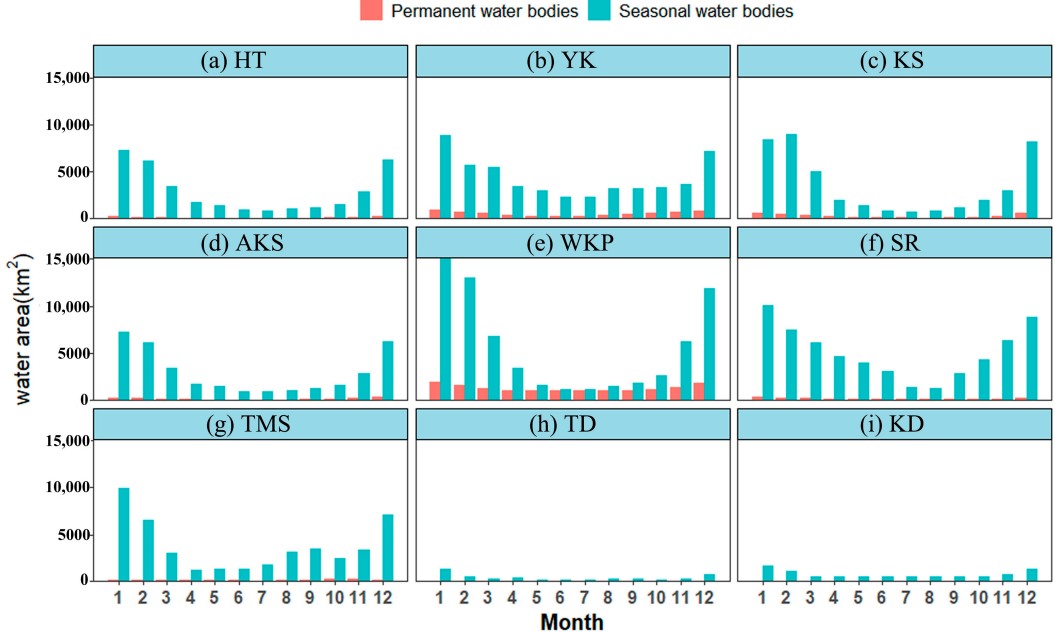

**Figure 8.** The monthly variation of surface water bodies for 1984 to 2019 in the (**a**) HT; (**b**)YK; (**c**) KS; (**d**) AKS; (**e**) WKP; (**f**) SR; (**g**) TMS; (**h**) TD; and (**i**) KD. HT, Hotan River Basin; YK, the Yarkant River Basin; KS, Kashgar River Basin; AKS, Aksu River Basin; WKP, Weikan-Kaidu-Peacock River Basin; SR, Rivers in Southern Xinjiang; TMS, the Tarim Main Stream; TD, Taklamakan Desert; and KD, Kumtag Desert.

*3.4. Yearly Changes in Open-Surface Water Bodies in the TRB*

Notably, because the Landsat 5 images from 1986 to 1991 did not fully cover the entire TRB, images for these six years were not used in calculating the annual change in open-surface water bodies in the TRB, and the analysis was calculated starting in 1992.

Over the last 28 years, the area of permanent water bodies increased slowly but not significantly (5.1 km$^2$/year, $p = 0.0735$), while seasonal and maximum water bodies experienced a sharp increase

of 744.0 and 754.7 km²/year, respectively, and the trend in both was extremely significant ($p < 0.001$). Regime shift detection revealed mutation points in three water body types: permanent water bodies did not produce mutations over 29 years, while both seasonal and maximum water bodies produced mutations in 2010 (Figure 9). The trends in permanent water bodies, seasonal water bodies, and maximum water bodies area in the nine sub-basins of the TRB were similar to the whole basin with an upward trend except for the permanent water bodies in AKS (−0.17 km²/year) and WKP (−8.16 km²/year), and the mutation points for the three types of the water bodies area in each sub-basin were not concentrated but were located within the period 2001–2013 (Figure 10 and Table 3).

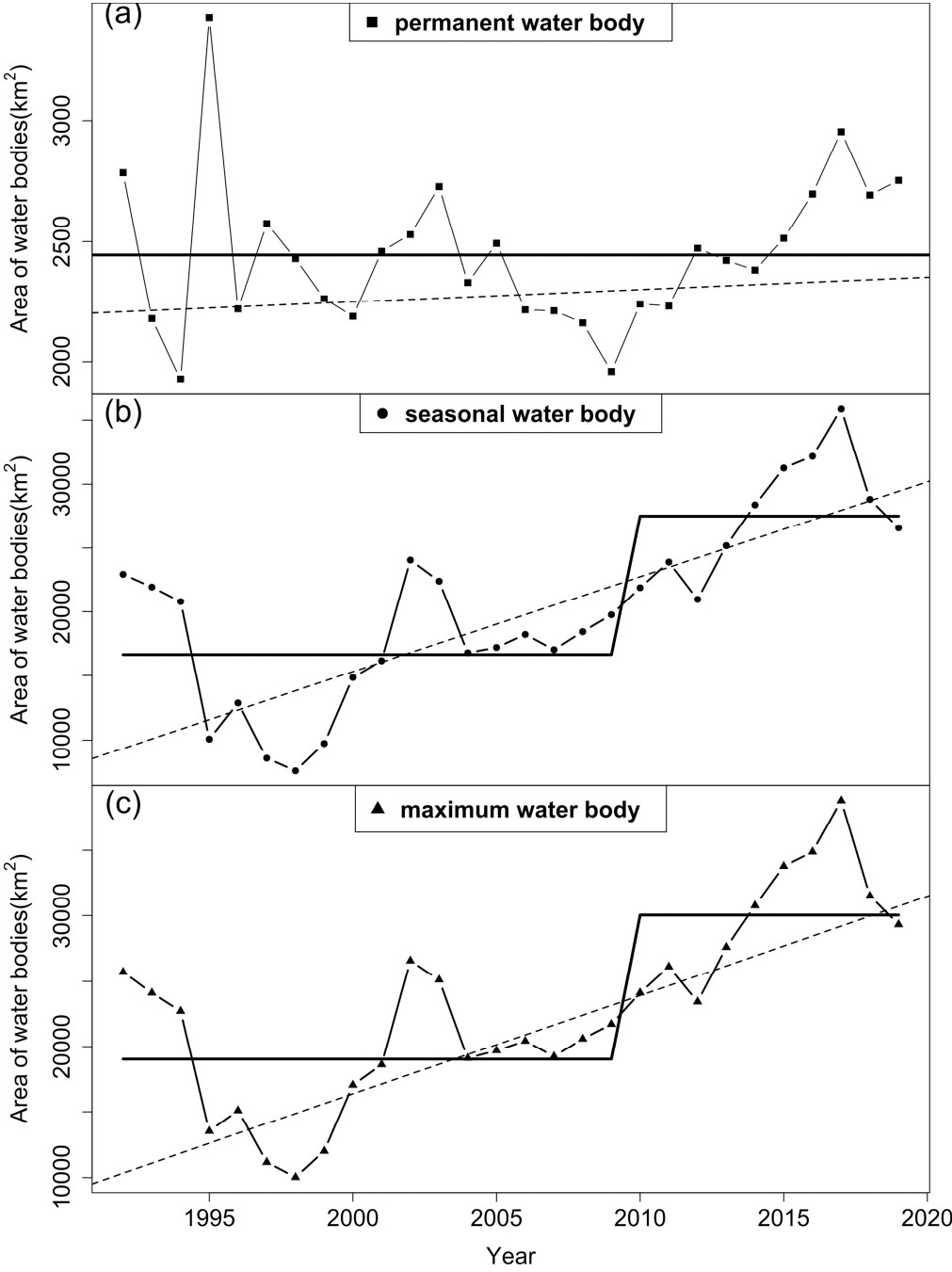

**Figure 9.** The inter-annual dynamics of water bodies areas during 1992–2019, with the Theil–Sen (TS) trends fitted in dashed lines and the average annual area regime shifts denoted in bold lines: (**a**) permanent water bodies; (**b**) seasonal water bodies; and (**c**) maximum water bodies.

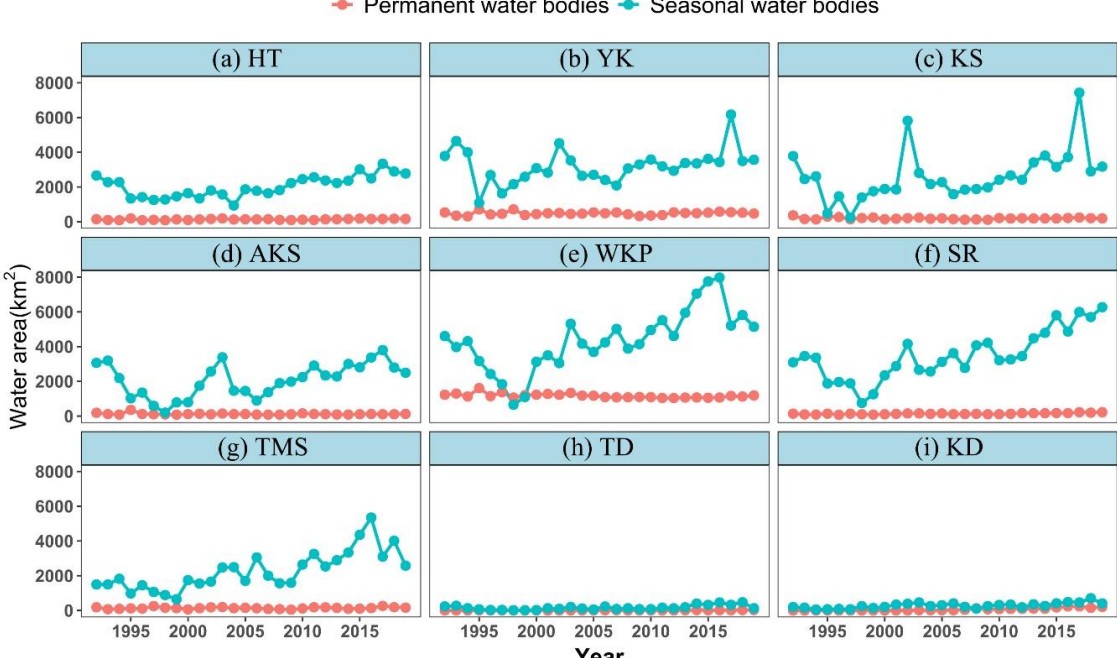

**Figure 10.** The annual variation of open-surface water bodies for 1992 to 2019 in the (**a**) HT; (**b**) YK; (**c**) KS; (**d**) AKS; (**e**) WKP; (**f**) SR; (**g**) TMS; (**h**) TD; and (**i**) KD. HT, Hotan River Basin; YK, the Yarkant River Basin; KS, Kashgar River Basin; AKS, Aksu River Basin; WKP, Weikan-Kaidu-Peacock River Basin; SR, Rivers in Southern Xinjiang; TMS, the Tarim Main Stream; TD, Taklamakan Desert; and KD, Kumtag Desert.

**Table 3.** Statistical summary of surface water bodies area dynamics in TRB's nine zones. HT, Hotan River Basin; YK, the Yarkant River Basin; KS, Kashgar River Basin; AKS, Aksu River Basin; WKP, Weikan-Kaidu-Peacock River Basin; SR, Rivers in Southern Xinjiang; TMS, the Tarim Main Stream; TD, Taklamakan Desert; and KD, Kumtag Desert.

| Zone | Water Body Type | Abrupt Point | Rate of Change (year$^{-1}$) | |
|---|---|---|---|---|
| | | | Area (km$^2$) | $p$-Value |
| HT | Permanent Water bodies | | 2.29 * | <0.001 |
| | Seasonal Water bodies | 2009 | 65.03 * | <0.001 |
| | Max Water bodies | 2009 | 65.75 | <0.001 |
| YK | Permanent Water bodies | | 3.07 | 0.0515 |
| | Seasonal Water bodies | | 46.35 * | 0.0024 |
| | Max Water bodies | | 54.77 * | 0.0016 |
| KS | Permanent Water bodies | | 0.70 | 0.759 |
| | Seasonal Water bodies | 2013 | 79.80 * | <0.001 |
| | Max Water bodies | 2013 | 74.34 * | <0.001 |
| AKS | Permanent Water bodies | | −0.17 | 0.493 |
| | Seasonal Water bodies | 2011 | 99.80 * | <0.001 |
| | Max Water bodies | 2011 | 100.15 * | <0.001 |
| WKP | Permanent Water bodies | 2006 | −8.16 * | <0.001 |
| | Seasonal Water bodies | 2003 | 147.95 * | <0.001 |
| | Max Water bodies | 2003 | 143.80 * | <0.001 |
| SR | Permanent Water bodies | 2012 | 3.467 * | <0.001 |
| | Seasonal Water bodies | 2008 | 155.06 * | <0.001 |
| | Max Water bodies | 2008 | 154.30 * | <0.001 |

**Table 3.** *Cont.*

| Zone | Water Body Type | Abrupt Point | Rate of Change (year$^{-1}$) | |
| --- | --- | --- | --- | --- |
| | | | Area (km$^2$) | *p*-Value |
| TMS | Permanent Water bodies | | 0.45 | 0.412 |
| | Seasonal Water bodies | 2003 | 96.98 * | <0.001 |
| | Max Water bodies | 2003 | 99.35 * | <0.001 |
| TD | Permanent Water bodies | 2013 | 0.33 * | <0.001 |
| | Seasonal Water bodies | | 7.37 * | <0.001 |
| | Max Water bodies | | 7.68 * | <0.001 |
| KD | Permanent Water bodies | 2008 | 7.47 * | <0.001 |
| | Seasonal Water bodies | 2001 | 12.02 * | <0.001 |
| | Max Water bodies | 2001 | 21.66 * | <0.001 |

\* indicates a significant value with 95% confidence interval.

To further study the interannual variation of surface water bodies, we show the conversion among permanent water bodies, seasonal water bodies, and non-water and the influence factors of open-surface water bodies in Sections 3.5 and 3.6, respectively.

### 3.5. Conversions of Open-Surface Water Bodies in the TRB

This study divided the entire study period (1986–2019) into three sub-periods (1986–1999, 2000–2010, and 2011–2019) to further demonstrate the conversion among permanent water bodies, seasonal water bodies, and non-water bodies in the TRB, the result is shown in Figure 11.

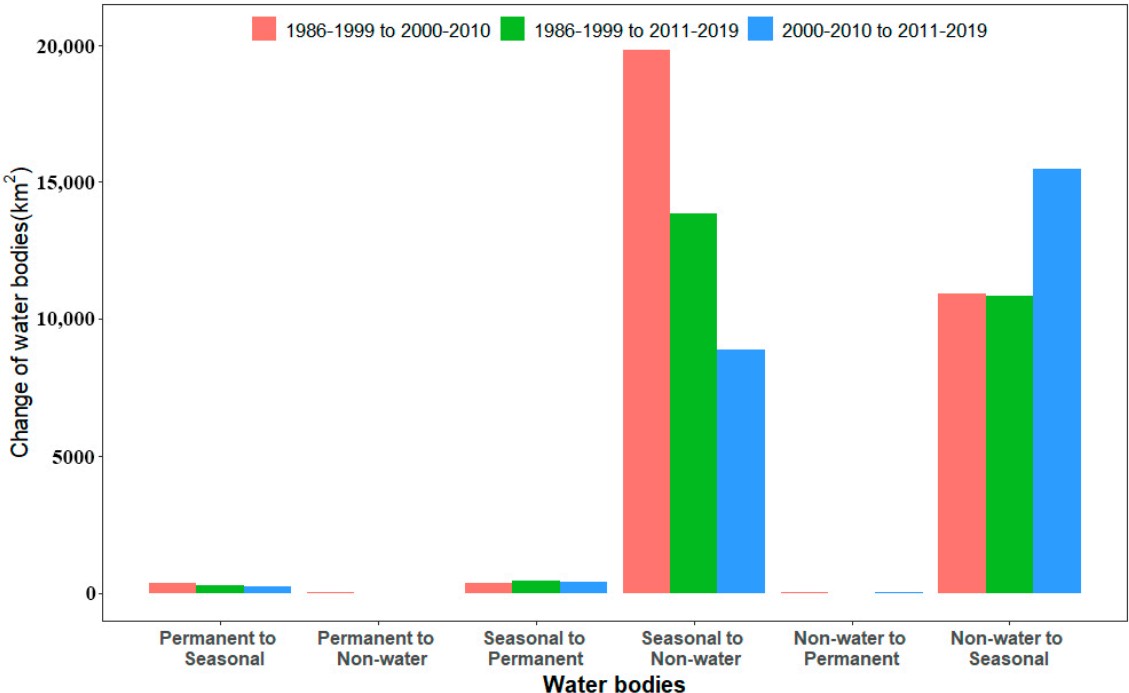

**Figure 11.** The conversion of open-surface water bodies in the TRB.

Between the first two stages (1986–1999 and 2000–2010), the change in water bodies type was mainly the mutual conversion between seasonal water bodies and non-water, with 20,476.64 km$^2$ of seasonal water bodies converted to non-water and 10,932.33 km$^2$ of non-water converted to seasonal water bodies and a decrease in seasonal water bodies and an increase in non-water. This was followed

by the mutual conversion between permanent and seasonal water bodies, with 345.27 km$^2$ of permanent water bodies converted to seasonal water bodies and 372.96 km$^2$ of seasonal water bodies converted to permanent water bodies and an increase in permanent water bodies and a decrease in seasonal water bodies. Finally, there was the mutual conversion between permanent and non-water, with only 18.46 km$^2$ of permanent water bodies converted to non-water and 11.62 km$^2$ of non-water converted to permanent water bodies. Between the latter two stages (2000–2010 and 2011–2019), the change in water bodies types continued to be dominated by the mutual conversion between seasonal and non-water, with 8876.55 km$^2$ of seasonal water bodies converted to non-water and 15,467.51 km$^2$ of non-water converted to seasonal water bodies and an increase in seasonal water bodies and a decrease non-water. This was followed by the mutual conversion between permanent and seasonal water bodies, with 249.12 km$^2$ of permanent water bodies converted to seasonal water bodies and 392.77 km$^2$ of seasonal water bodies converted to permanent water bodies and an increase in permanent water bodies and a decrease in seasonal water bodies. Finally, there was the mutual conversion between permanent and non-water, with only 2.29 km$^2$ of permanent water bodies converted to non-water and 11.74 km$^2$ of non-water converted to permanent water bodies. The overall (from 1986–1999 to 2011–2019) change in water bodies type is mainly the mutual conversion between seasonal and non-water, with 13,849.46 km$^2$ of seasonal water bodies converted to non-water bodies and 10,827.56 km$^2$ of non-water converted to seasonal water bodies and a decrease in seasonal water bodies and an increase in non-water. This was followed by the mutual conversion between permanent and seasonal water bodies, with 289.72 km$^2$ of permanent water bodies converted to seasonal water bodies and 454.37 km$^2$ of seasonal water bodies converted to permanent water bodies and an increase in permanent water bodies and a decrease in the area of seasonal water bodies. Finally, there was the mutual conversion between permanent and non-water, with only 7.14 km$^2$ of non-water converted to permanent water bodies and no permanent water bodies converted to non-water.

During 1986–1999, permanent water bodies totaled 2004.021 km$^2$, seasonal water bodies totaled 49,741.98 km$^2$, and non-water totaled 65,258.19 km$^2$. During 2000–2010, permanent water bodies totaled 2024.861 km$^2$, seasonal water bodies totaled 40,169.98 km$^2$, and non-water totaled 74,809.35 km$^2$. During 2011–2019, permanent water bodies totaled 2091.214 km$^2$, seasonal water bodies totaled 46,585.29 km$^2$, and non-water totaled 68,327.5 km$^2$. It can be seen that the area of permanent water bodies has been increasing slightly during the three stages. Seasonal water bodies have experienced a trend of decreasing and then increasing. On the contrary, non-water has experienced a trend of increasing and then decreasing.

Ran [76] found that, during the 25 years from 1990 to 2015, the main changes occurred among water bodies, unused land, and grassland, while the transformation of water bodies from cropland, forest, and residential land is relatively small. This showed that, during the entire period from 1986–1999 to 2011–2019, human activities were not the main reason for the mutual transformation between seasonal water bodies and non-water. We found that the conversion of non-water into seasonal water bodies mainly occurred in the western mountainous where glaciers are distributed over a large area. This may be due to the increase in temperature in the past 30 years, which has accelerated the melting of snow and ice in the mountains (Table 4). In recent years, there have been many studies on TRB's glaciers [77–81]. Affected by topography, climate, glacier structure, and other factors, TRB's glaciers have obvious regional differences, but the overall trend is to shrink. The meltwater covered the non-water near the glacier and forms water bodies. The transformation of water bodies into non-water mainly occurred in the inner plains of the basin. This may also because of the increase in temperature, which led to strong evaporation, and part of the seasonal water bodies was transformed into non-water.

**Table 4.** Changes of typical glaciers in the TRB during recent decades.

| Typical Glaciers | Time Interval | Area Change (km$^2$) | Rate of Change (%) | Annual Change (%·year$^{-1}$) | Changes in Ice Reserves (km$^3$) |
|---|---|---|---|---|---|
| Glacier No. 72, Qingbingtan, Tomur Peak [79] | 1964–2009 | −1.53 | −14.7 | −0.03 | −0.0141 |
| Kunlun Mountains [80] | 1976–2011 | −1243.6 | −12 | −0.34 | — |
| West Kunlun Peak District [80] | 1990–2011 | −16.83 | −0.62 | −0.03 | — |
| West Kunlun Mountains [81] | 1977–2013 | −91.12 | −2.95 | −0.08 | −20.21 |
| Qogir North Slope Glacier [78] | 1978–2014 | −53.37 | −6.81 | −0.19 | — |
| Kelechin River Basin [77] | 1978–2015 | −145.78 | −8.00 | −0.22 | — |

Similar to the overall trend from 1986–1999 to 2000–2010, the area of permanent water bodies increased slowly, the area of seasonal water bodies declined, and the area of non-water increased. The changes mainly occurred between seasonal water bodies and non-water. Human activities at this stage were not the main factors affecting the transformation of surface types. Rising temperature caused the conversion of non-water in mountainous into water bodies, and the conversion of seasonal water bodies in plains to non-water was through melting glaciers and enhanced evaporation.

From 2000–2010 to 2011–2019, the area of permanent water bodies increased slowly, seasonal water bodies rose rapidly, and the area of non-water bodies declined. Changes still happened mainly between seasonal water bodies and non-water. Human activities at this stage were still not the main factor affecting the transformation of land types. Rising temperature caused the non-water in mountainous into water bodies, and the plain seasonal water bodies were transformed into non-water by melting glaciers and enhanced evaporation respectively. However, the increase in precipitation during this period has made up for the evaporation of plain areas. Therefore, seasonal water bodies have risen rapidly, while non-water bodies have declined.

*3.6. Relationship between the Climatic Factors and Yearly Maximum Water Bodies*

From 1992 to 2019, the area of TRB's maximum water body area showed an upward trend. During this period, we calculated the Pearson correlation coefficients between the maximum water body area and annual precipitation, annual temperature, and annual evapotranspiration. On the interannual scale, the maximum water area of the TRB is significantly correlated with precipitation, annual average temperature, and annual evapotranspiration ($p < 0.001$, $p = 0.0019$, and $p = 0.0023$, respectively) (Figure 12). The highest correlation coefficient is annual precipitation, followed by temperature and evapotranspiration (0.653 > 0.649 > 0.552) (Figure 12).

It is well understood that the increase in precipitation directly affects the area of water bodies. The increase in temperature also expands the area of water bodies by accelerating the melting of ice and snow. Therefore, both have a significant positive correlation with water bodies. However, we were surprised to find that evapotranspiration is also significantly positively correlated with water bodies. The influencing factors of evapotranspiration in TRB include solar radiation, soil heat flux, air temperature, humidity, air pressure, air resistance, etc. [82]. According to that, the increase in evapotranspiration can be explained from two aspects: (1) the expansion of the water body area provides more underlying surfaces that are easy to evaporate [82]; and (2) rising temperature provides a sufficient source of heat for evapotranspiration. Moreover, the increase in evapotranspiration did not offset the positive effects of precipitation and melted ice and snow on the expansion of the water area. The combination of all conditions resulted in a positive correlation between the water area and evapotranspiration.

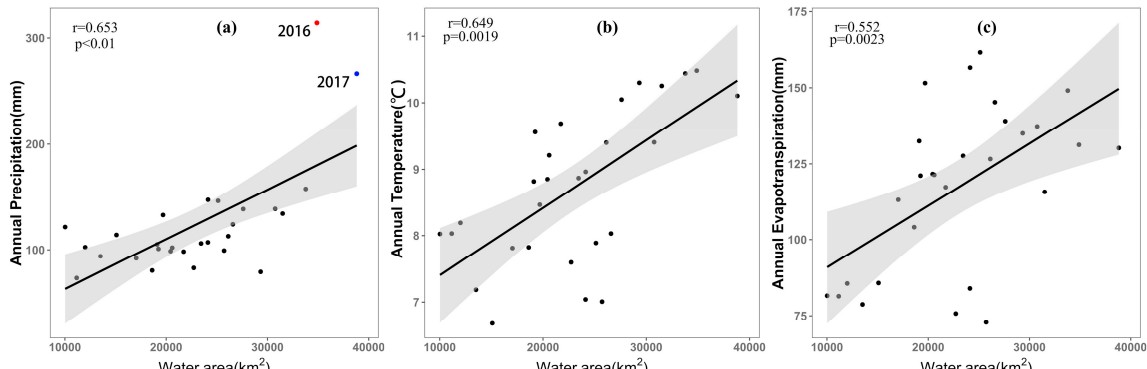

**Figure 12.** The Pearson correlation coefficient between water area and (**a**) annual precipitation (The red dot represents 2016 and the blue dot represents 2017), (**b**) annual temperature, and (**c**) annual evapotranspiration from 1992–2019. The black lines in the figure are linear regression curves. The gray area represents the 95% confidence interval.

In Figure 12a, there are two very significant outliers (marked by red (2016) and blue (2017)). In 2016, there was the largest precipitation in 29 years (314.2 mm), but the maximum water area appeared in 2017. This may be caused by two reasons: (1) There could be a time lag from precipitation to runoff and convergence into surface water bodies. When studying the relationship between the water area and precipitation of Poyang Lake, Wang et al. [35] found that there was a 1–2-month lag between the change in water area and precipitation. Therefore, the maximum of water area in the TRB will be slightly later than the maximum rainfall. (2) To reduce the uncertainty of the image data quality mark and the potential errors caused by other small-probability problems in image preprocessing, this study set water bodies with WIF < 1% as non-water. Therefore, in 2016, a large area of short-lived temporary water bodies occurred due to a large amount of precipitation, but, for most pixels, their WIF was less than 1% so they were classified as non-water, resulting in the surface water area not reaching the maximum value with precipitation.

## 4. Discussion

### 4.1. Comparison with JRC Yearly Water Classification History

In this section, we compare our results with JRC Yearly Water Classification History (v1.1) [8], a dataset generated from 3865,618 scenes in Landsat 5, 7, and 8 images covering the global spatiotemporal distribution of surface water bodies during 1984–2018. The dataset classifies each pixel into two categories, namely water bodies and non-water, where water bodies are divided into permanent water bodies and seasonal water bodies based on whether they are identified each month, the sum of which is considered the maximum water bodies area of the JRC data. The results are shown in Figure 13. Overall, the spatial distribution pattern and temporal trends of surface water bodies were similar between the two datasets, and the permanent water bodies in both this study and JRC have changed little over time. However, there was a big difference between the two: the multi-year average of permanent water bodies based on JRC data was 9471.44 km$^2$, 7027.5 km$^2$ more than in this study, and the multi-year average of seasonal water bodies based on JRC data was 26,647.7 km$^2$, 6161.9 km$^2$ more than in this study. This is because the JRC defines permanent water bodies and seasonal water bodies differently from this study. JRC defines permanent and seasonal water bodies according to the number of months that are identified as water bodies in a year, considered as permanent water bodies if all 12 months are identified as water bodies, and considered as seasonal water bodies if the number of months that are identified as water bodies falls within the 1–11 interval. Thus, the JRC results are larger than this study.

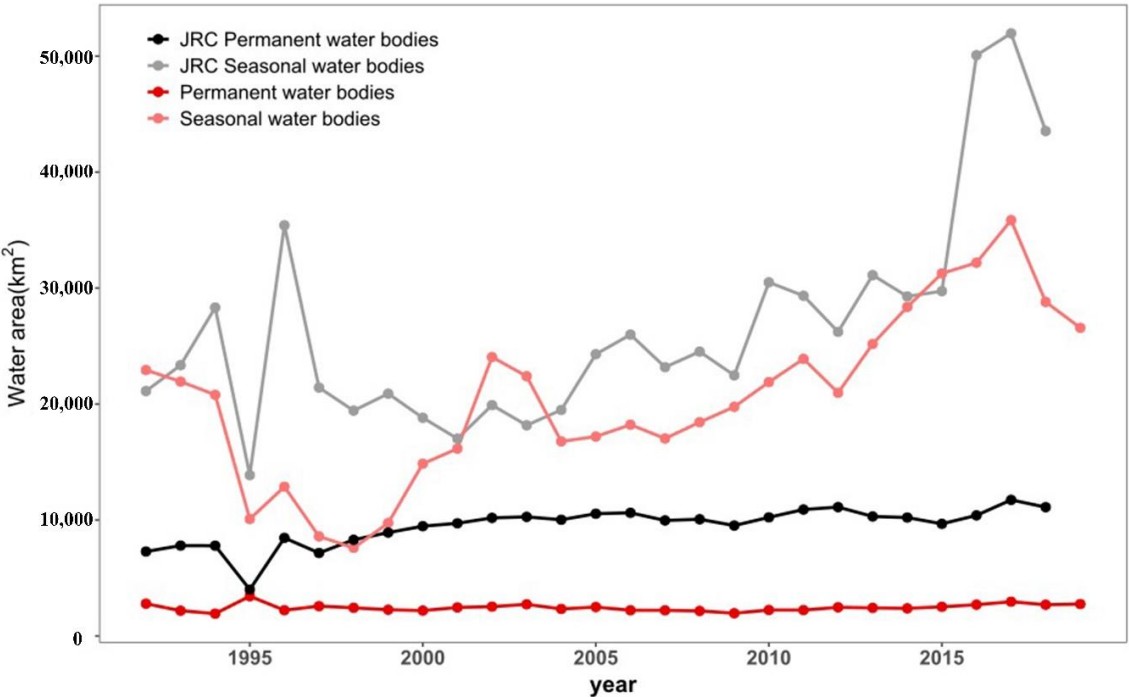

**Figure 13.** Comparison of TRB's yearly water bodies area with JRC data.

*4.2. Attribution of Open-Surface Water Bodies Changes in the TRB*

There is a good correlation between the maximum water bodies area in the TRB and precipitation. The results indicate that precipitation is an important factor in controlling the changes of open-surface water bodies in the TRB. In addition, the entire TRB showed a trend of slightly increasing precipitation from 1992 to 2019; in particular, in 2016, against the backdrop of a super-strong El Niño event [83], the average precipitation in the TRB in 2016 reached 314.2 mm [63] (Figure 12), the highest since 1961. The increase in precipitation has been accompanied by an increasing trend of permanent, seasonal water bodies in the TRB from 1992 to 2019.

Moreover, there is no doubt that precipitation is not the only climatic factor that influences changes in open-surface water bodies in the TRB. Open-surface water bodies in the TRB showed a clear "single valley" trend, with a sharp decline from January to April, a slow decline from April to July, a slow rise from July to October, and a sharp rise from October to December. This is because temperature controls the phenological effect, snowmelt, and evaporation, which together cause the open-surface water bodies of the TRB to show obvious intra-year differences. In this study, we observed an upward trend in the correlation coefficient between the maximum water bodies and yearly precipitation from 1992 to 2019. The likely reason is that the impact of human activities on the TRB has decreased over the past few years. Many studies have demonstrated the effects of human activities on surface water bodies [54,84]. The TRB has experienced intense human activity over the years, with complex impacts on open-surface water bodies. On the one hand, over the past 60 years, the TRB has been subjected to intensive human economic and social activities centered on the exploitation and use of water resources, and, as a result of agricultural irrigation, grazing, urban expansion, and the construction of inappropriate water facilities such as the Daxihaizi Reservoir, open-surface water bodies have been encroached upon by other land-use types, the lower Tarim River has been completely cut off, and the tail of the river—Lop Nor and Lake Tetema—dried up in 1970 and 1972, respectively, resulting in a rapid decline in open-surface water bodies. However, human activities can also increase the area of surface water bodies. In KD, for example, the region experienced a significant increase in permanent water bodies as a result of the construction of the new Lop Nor Potash Plant. Comparing the open-surface water bodies in KD during 1986–2000 and 2001–2019 (before and after the construction of the Lop Nor Potash Plant), it was

found that the permanent water bodies in KD increased by 1.52 km$^2$ and the seasonal water bodies increased by 372.65 km$^2$; in addition, due to the implementation of the ecological water transfer project to the lower Tarim River in 2000, the water bodies of Lake Tetema, which had been dried up since 1972, were restored; comparing the open-surface water bodies in the Tetema Lake region during 1986–2000 and 2000–2019 (before and after the ecological water transfer), it was found that the permanent water bodies in the Tetema Lake region increased by 2.31 km$^2$ and the seasonal water bodies increased by 616.9 km$^2$, indicating the great influence of human activities on the increase of surface water bodies (Figure 14).

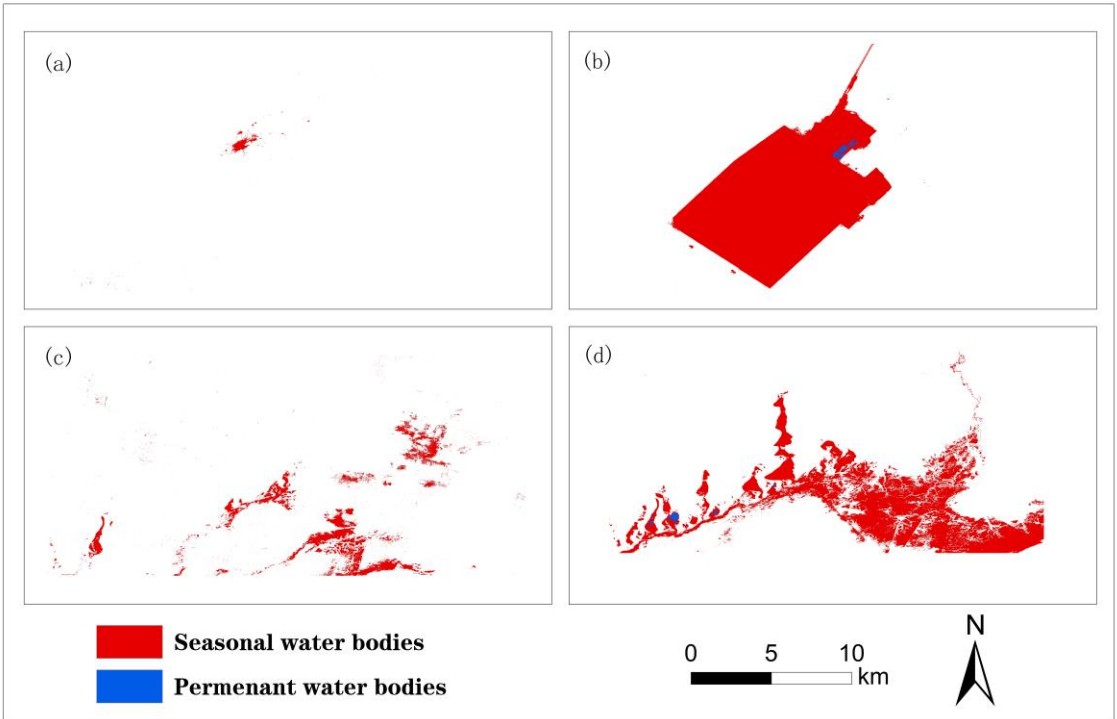

**Figure 14.** Open-surface water bodies: (**a**) before (1986–2000) and (**b**) after (2001–2019) the Lop Nor Potash Plant; and (**c**) before (1986–2000) and (**d**) after (2001–2019) the ecological water delivery.

In short, precipitation is the main factor that affects the interannual variability of the water body area, while temperature controls the intra-year variation in TRB. Human activities have positive and negative effects on the change of water body area, but it is limited to local areas and not the main factor.

*4.3. Advantages and Uncertainties of This Study*

In previous studies, water body detection was performed by setting constant thresholds for water indices, which can be subjective, time-consuming [39], and difficult to extrapolate to other regions due to differences in images and locations [85]. Zou et al. [52] did not use fixed thresholds but performed water body detection based on a combined MNDWI and EVI/NDVI relationship, which can well distinguish vegetation from water bodies and perform well in heavily vegetated areas but not in unvegetated bare ground. Therefore, based on these two types of water detection ideas, a new water bodies identification rule (ARWDI) for arid regions with sparse vegetation cover was established in this study by setting different thresholds (−0.1 and 0.1) for the two water indices of NDWI and MNDWI, combined with the relationship of water index and vegetation index.

Most of the water bodies indices have more omission errors than commission errors [39,40]. The mixed pixels at the edges of water bodies may be the major reason for water pixel omission [40]. In contrast, Zou et al.'s [52] detection rule has a greater commission error than omission error, with many non-water pixels identified as water. The classification errors in this study were mainly due to

commission error, with 98 water samples identified as non-water and 41 non-water samples identified as water bodies. Low albedo surfaces, including shadows of mountains, buildings, and clouds, are a major source of commission errors in water detection [39,86]. Although cloud quality bands are used for masking clouds in data preprocessing, undetected residual clouds and cloud shadows can still cause commission errors. In this study, the water frequency threshold (0.01) was used to remove most of the noise from the water frequency map. However, along with the noise removal, the frequency threshold also removes some temporary water signals, which may lead to an underestimation of the water bodies area.

## 5. Conclusions

In this study, we proposed a surface water body identification rule, ARWDR, which is applicable to arid areas. ARWDR effectively solves the problem that Zou et al.'s [52] rule cannot effectively distinguish between bare soil and water bodies. Then, we investigated the long-term changes of open-surface water bodies in the TRB from 1992 to 2019 based on all available Landsat 5, 7, and 8 images in the GEE platform by ARWDR. The open-surface water bodies were classified into permanent water bodies and seasonal water bodies according to the inundation frequency. The spatial distribution and monthly and annual changes of permanent water bodies and seasonal water bodies in the TRB over the past 28 years were calculated. Overall, this study provides the following conclusions:

(1) The distribution of surface water bodies in the TRB shows obvious spatial heterogeneity. The water bodies are mainly distributed in mountainous areas and piedmont plains, and there are almost no permanent water bodies in the basin.

(2) Phenological effects and snowmelt and evaporation, which are affected by temperature changes, together cause the surface water bodies of the TRB to show obvious intra-year differences, that is decreasing from January to July, and then increasing to December.

(3) From 1992 to 2019, with the increase of precipitation, the implementation of ecological water transportation, and other measures, the permanent water bodies and seasonal water bodies of the TRB showed an increasing trend.

The results are meaningful as a further reference for watershed management of TRB in the context of climate change to ensure the security of water resources. However, this study did not thoroughly investigate the effects of human activities on the long-term changes of TRB's open-surface water bodies. In the next work, we will use more economic and social data (such as population, GDP, cultivated field area, and irrigation water consumption) to investigate the changes and explore the reasons.

**Supplementary Materials:** The following are available online at http://www.mdpi.com/2073-4441/12/10/2822/s1, Figure S1: Dynamic maps of the historical changes of six typical water bodies. (**a**) the source of the mainstream of the Tarim River, (**b**) Bosten Lake, (**c**) Daxihaizi Reservoir, (**d**)Yarkant River, (**e**) Tetema Lake, and (**f**) Lop Nor Potash Plant.

**Author Contributions:** Conceptualization, J.C. and Y.G.; methodology, J.C. and J.B.; software, J.C. and S.Y.; validation, J.C., T.K., and Y.G.; formal analysis, J.C.; investigation, J.C., T.K., J.B., and Y.G.; resources, J.C., T.K., and Y.G.; data curation, J.C. and Y.G.; writing—original draft preparation, J.C.; writing—review and editing, J.C., T.K., S.Y., J.B., and Y.G.; visualization, J.C. and K.C.; supervision, Y.G.; project administration, Y.G.; and funding acquisition, Y.G. All authors have read and agreed to the published version of the manuscript.

**Funding:** This study was funded by the Science and Technology Service Network Initiative Project of the Chinese Academy of Sciences, grant number KFJ-STS-ZDTP-036 and Research on Climatic and Ecological Changes and Resource and Environmental Carrying Capacity Enhancement in Xinjiang.

**Acknowledgments:** This study was supported by the Science and Technology Service Network Initiative Project of Chinese Academy of Sciences (No. KFJ-STS-ZDTP-036). Lastly, the authors would like to acknowledge the anonymous reviewers and editors whose thoughtful comments helped to improve this manuscript

**Conflicts of Interest:** The authors declare no conflict of interest.

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
