# Peer review of "Open-Surface Water Bodies Dynamics Analysis in the Tarim River Basin (North-Western China), Based on Google Earth Engine Cloud Platform"

_water, doi:10.3390/w12102822_

Round 1

Reviewer 1 Report

- lines 2-4 - the title should be adjusted under the form of ”Open-surface Water Bodies in Plan Dynamics Analysis in the Tarim River Basin (North-Western China), Based on Google Earth Engine Cloud Platform”;

- lines 13-30 - the abstract should be revised: ”water” after ”water”, ”previous study”???????????????? - which??, ”epoch scale” ..., ”visual interpretation” - which scale???, or the term refers to digital algorithms, what type of ”56,284 scenes” - true colour???, ”(4) The changes in open-surface water bodies in the TRB were related to climate change and the intensity of human activities.” - this should be proved using serious statistic components;

- lines 40-41 - ”Due to climate change and human activities, permanent water bodies have been reduced by nearly 90,000 km2 from 1984 to 2015” ... an effective official statistics should be considered;

- lines 123-140 - you should reorganise this text, eventually under a systematic structure with multiple phrases from the left margin; the reader makes important efforts to read-it;

- lines 140-141 - also, please, reorganise the Figure 1, with modified technical components: another colour range - ex. from dark brown to moderate green (attached); clarified study area lines - eventually, put there all the studied sub-basins with symbols or codes (HT, YK ... or 1, 2, 3 ...) and not covering the rivers network; please do not create a confusion between watersheds/sub-basins and morphological units; each of them can be placed on a well organised figure;

- lines 244-245 - please, include some more details about ”Comparing the eleven methods with the samples obtained by visual interpretation, the overall accuracy and kappa coefficients of the different methods”; did you applied all these Water Indexes in several pilot/representative areas for comparison? ... etc.;

- line 250 and more (3.2. - 3.5 sub-chapters) - regarding spatial distribution and evolution of open-surface water bodies in the TRB, did you established which/how of these water bodies represents more than a hydric spot?; are those water bodies having any potential regarding the use by anthropic (water resource) or natural (wetlands) environments?; being in arid area, with high rates of mineralization grace of evapotranspiration, is there any signification to be identified in any direction?;

- sub-chapter 3.6. a graphical correlation (not in hydrograph system) Max. Wat. Bodies = f(Precip., or ToC and Ev) should be very enlightening;

- generally, did the authors validated their virtually work using several pilot/direct study case areas?; along several last years of observed period, using GPS and other disposable technics, did they verified the real existence of these open-surface water bodies and their spatial extension to be compared with results from digital extraction?; a superpose of several polygons obtained from GPS measurements over digital grids would have been very useful and strong evidences for validation;

- validate using other scientific productions or theoretically is a half one for such an extended and vulnerable area regarding water resources;

- the practical use of the study has a great potential (ex. sub-chapter 4.2), or by identifying the stable and useful water bodies, or abandoning the search of water in vulnerable/arid areas and indicating other technical solutions (ex. water transfer from contiguous mountainous area - a composite and technical map, that make possible to reconsidering the Tarim River Basin water resources).

Reviewer 2 Report

It would be good to check spelling – for example word rose on the line 394 should be rise I think.

Reviewer 3 Report

Reject

Overall approaches are sound, but only minimal discussions are available to discuss the results. In the results section, the authors did not discuss why the outcomes occurred, and they just stated the values of the results. Also, the discussion section does not cover those flaws. I think an intense scientific discussion based on the results is required to be published in this journal. Please see the specific comments below.

- Line 38: Remove ‘However.’

- Line 41: Please provide a percentage decrease for 90,000 sq. km and it can highlight the severity.

- Line 116 to 140: Please add proper references for the information.

- Figure 1. Please provide spatial maps of precipitation and temperature, and explain the characteristics of them for international readers. Also, please use a more distinct color for the names of the sub-basins. Some of them were hard to distinguish. 

- Figure 2. Why there was a steep decline in 2012? Please discuss. 

- Line 168: The authors need to explain some specific descriptions about the water detection rules of Zou et al. with a separate section; it may help to understand why the detection rules did not perform well in the TRB.

- Line 184 to 186: The authors need to explain more how those thresholds were found. Any statistical background or specific approaches? Please address.

- Line 246 to 247: Why the overall accuracy and kappa coefficient ARWDR are the highest? Why the other methods showed lower accuracy (e.g., EWI, LSWI+VI)? The authors need to address this, and it may highlight why the authors have used the ARWDR method.

- Line 251 to 259: Not just stated the numbers of permanent water bodies, please describe why some regions showed higher or lower permanent water bodies with their hydro-climatic conditions. For example, are the Tetema Lake and Lop Nor Potash Plant regions affected by monsoonal climate conditions compared to the others?

- Table 2. Again, Table 2 has lots of information. Please explain and discuss why some regions showed higher or lower permanent water bodies with their hydro-climatic conditions.

- Table 2. Please provide the full names of the abbreviations in the table caption (e.g., HT, YK)

- Line 266 to 280: Please discuss why the seasonal variations have occurred. Precipitation variation?

- Figure 8: It is better to add the names of each sub-region on the bar charts.

- Figure 10: Again, why there were overall increases in seasonal water bodies? Increases in precipitation or snowmelt? Please explain and discuss it with related factors.

- Figure 11. Not just stated the numbers, and please add scientific discussions about the results. (e.g., why those changes have occurred, impacts of the changes on water resources, agriculture, or water-related issues)

- Line 343 to 353: Which statistical test the authors used? Linear regression analysis? Please add the details. Also, scatter plots may show the correlations better.  

- Table 4. Again, do not just state the values, please add proper discussions based on the results. 

- Figure 12 (a). There was a steep increase in precipitation in 2016, but the water area did not follow the increase. The authors need to address this point.

- The abstract and conclusion should be rewritten after revision.

Round 2

Reviewer 3 Report

Thanks for addressing all of my comments, and the revised manuscript was improved. However, some minor and major issues should be addressed to be published.

Minor comment
- Figure 1: Please provide full names for the abbreviation in the figure caption. Also, please provide more prominent and high-resolution legends. The current forms are hard to see.
- Figure 2. The revised version can highlight why there was a steep decrease in 2012. However, please add an explanation for the Landsat status in the revised manuscript, not just add the figure's statements.
- Line 207: because -> because of

Major comments
- 3. Results: In the revised manuscript, the authors pointed out an “increase in temperature in the past 30 years, which has accelerated the melting of snow and ice in the mountains.” Can you prove the increases in the snow and glaciers melt from the remote sensing data or other proper sources? This point is the most crucial reason; thus, it should be validated with appropriate analysis. Please provide some figures or tables if needed.
- Conclusion: Not just summarizing the results, please summarize the scientific contributions and shortcomings of this study, and mention further required studies.

Round 3

Reviewer 3 Report

I think the authors have made revisions I suggested. Therefore, I have no other questions but moderate English editing is required.